# Idiopathic Intracranial Hypertension: Evaluation of Admissions and Emergency Readmissions through the Hospital Episode Statistic Dataset between 2002–2020

**DOI:** 10.3390/life11050417

**Published:** 2021-05-05

**Authors:** Susan P. Mollan, Jemma Mytton, Georgios Tsermoulas, Alex J. Sinclair

**Affiliations:** 1Birmingham Neuro-Ophthalmology, University Hospitals Birmingham NHS Foundation Trust, Birmingham B15 2TH, UK; 2Health Informatics, University Hospitals Birmingham NHS Foundation Trust, Birmingham B15 2TH, UK; jemma.mytton@uhb.nhs.uk; 3Department of Neurosurgery, Queen Elizabeth Hospital, University Hospitals Birmingham NHS Foundation Trust, Birmingham B15 2TH, UK; georgios.tsermoulas@uhb.nhs.uk; 4Institute of Metabolism and Systems Research, University of Birmingham, Edgbaston, Birmingham B15 2TT, UK; 5Metabolic Neurology, Institute of Metabolism and Systems Research, University of Birmingham, Edgbaston B15 2TT, UK; a.b.sinclair@bham.ac.uk; 6Centre for Endocrinology, Diabetes and Metabolism, Birmingham Health Partners, Birmingham B15 2TH, UK; 7Department of Neurology, University Hospitals Birmingham NHS Foundation Trust, Birmingham B15 2TH, UK

**Keywords:** emergency, hospital episode statistic, idiopathic intracranial hypertension, neurosurgery, pseudotumor cerebri, papilloedema, shunt, stent, 30 days readmission, revision

## Abstract

With increasing incidence and prevalence of Idiopathic intracranial hypertension in the UK, the aim of this study was to explore emerging themes in Idiopathic intracranial hypertension using the Hospital Episode Statistics dataset and to quantify recent change in hospital admissions and surgeries performed within England. Methods: Hospital Episode Statistics national data was extracted between 1 April 2002 and 31 March 2019, and followed up until 31 March 2020. All those within England with a diagnosis of Idiopathic Intracranial Hypertension were included. Those with secondary causes of raised intracranial pressure such as tumors, hydrocephalus and cerebral venous sinus thrombosis were excluded. Results: 28,794 new IIH cases were diagnosed between 1 January 2002 and 31 December 2019. Incidence rose between 2002 to 2019 from 1.8 to 5.2 per 100,000 in the general population. Peak incidence occurred in females aged 25–29 years. Neurosurgical shunt was the commonest procedure performed (6.4%), followed by neovascular venous sinus stenting (1%), bariatric surgery (0.8%) and optic nerve sheath fenestration (0.5%). The portion of the total IIH population requiring a shunt fell from 10.8% in 2002/2003 to 2.46% in 2018/2019. The portion of the total IIH population requiring shunt revision also reduced over time from 4.84% in 2002/2003 to 0.44% in 2018/2019. The mean 30 days emergency readmissions for primary shunt, revision of shunt, bariatric surgery, neurovascular stent, and optic nerve sheath fenestration was 23.1%, 23.7%, 10.6%, 10.0% and 9.74%, respectively. There was a peak 30 days readmission rate following primary shunt in 2018/2019 of 41%. Recording of severe visual impairment fell to an all-time low of 1.38% in 2018/19. Conclusions: Increased awareness of the condition, specialist surgery and expert guidance may be changing admissions and surgical trends in IIH. The high 30 readmission following primary shunt surgery for IIH requires further investigation.

## 1. Introduction

Idiopathic Intracranial Hypertension (IIH) is characterized by an elevation of intracranial pressure (ICP) with no identifiable cause [1,2]. There is a rising incidence in this disease, and it appears that the incidence is related to country specific prevalence of obesity [3,4,5]. Incidence data reported 4.69 per 100,000 in the general population of England, United Kingdom (UK) were diagnosed with IIH in 2016 [3]. There was a prevalence in 2015 of 68 per 100,000 for women in the UK [4]. Despite the relative rarity of IIH the multidisciplinary manifestations of the condition lead these patients to access hospital care though a large number of hospital specialties. Headache is the predominant morbidity in over 90% [2,6] and persistent post-IIH headache a challenge to manage [7,8]. Sight loss is not uncommon as typically 7% will require surgical intervention to save vision [3,9]. Weight loss is known to induce remission of the disease; however, the formal tiered care weight management pathway within the National Health Service (NHS) takes time and has restrictions on access. For example, IIH is not currently classified as an obesity related illness, where other obesity related conditions qualify from a lower body mass index entry point for hospital based weight management services [10].

New insights into the underlying pathophysiology have been discovered, including the role of the glucagon-like peptide 1 (GLP-1) receptor agonist in cerebrospinal fluid (CSF) secretion and intracranial pressure regulation, and the role of androgens in CSF dysregulation in IIH [11,12,13,14,15]. These have been translated into clinical trials, with the first phase 2 trial of a novel treatment recently reported [16]. Clinical trials evaluating lifestyle approaches to weight loss methods are guiding clinical care [17,18], and a recent evaluation of Calcitonin Gene-Related Peptide (CGRP) migraine therapy is the first prospective study to consider headache therapy for persistent post-IIH headache [19]. Whilst progress for the condition is clear to chart, translation of trial evidence into routine clinical practice requires validation and funding, all of which takes time.

The 2015 Cochrane review identified the requirement for more research for treatment of IIH [20] and in the same year the neurology community, through the Association of British Neurologists, proposed a guideline for investigation and management of IIH [21,22,23]. The European Headache Federation commissioned a management guideline of IIH, in response to member’s requests [24]. Key messages within these guidelines recommended a reduction in repeat admissions for serial lumbar punctures; that surgery should only be indicated for visual dysfunction; and improvement in access to medical headache management was needed [23,24]. In 2017 a priority setting partnership was convened to bring the UK medical and patient communities together with the aim to set research objectives for IIH. This was accomplished by multiple rounds of surveys exploring care and aspirations for IIH [25]. 

In the UK activity from bench to clinical practice firmly brought IIH to the immediate attention of general neurologists, ophthalmologists and neurosurgeons. The aim of this study was to explore shifting themes in IIH using the Hospital Episode Statistics (HES) dataset and to quantify recent trends in hospital practice within England given the increased dissemination of best practice in IIH management at clinical meetings and in the literature, and increased disease visibility through surveys. 

## 2. Materials and Methods

### 2.1. Study Design and Setting

This study was conducted by means of a registered national data set, including all patients with IIH admitted for hospital care in England between 1 January 2002 and 31 December 2019. Data were obtained from the Hospital Episode Statistics (HES), an administrative dataset covering all National Health Service (NHS) Trusts in England, which processed over 125 million admitted patient, outpatient and accident and emergency records each year, generating a log of each clinical episode taking place in NHS hospitals or NHS commissioned activity in the independent sector. This includes accident and emergency attendances, admitted hospital care (inpatient activity) and ambulatory care admissions (day case admissions). University Hospitals Birmingham National Health Service (NHS) Foundation Trust, United Kingdom approved the study as service evaluation (audit) (Registered Code, Clinical Audit Registration and Management System: CARMS-15969).

Each record was anonymized and comprised specific demographic details of the admitted patient including age group, gender, ethnicity and geographical information such as location of treatment and domicile. University Hospitals Birmingham NHS Foundation Trust possesses a Data Re-Use Agreement for the interrogation of the HES. The research involved non-identifiable information, previously collected in the course of patient care and available for public use. Where there are less than five people in any category the result is not available to ensure anonymization is upheld.

A diagnosis of IIH will be made by the hospital specialist. Typically in the UK hospital specialists should follow the guidelines for this diagnosis which include the presence of papilloedema; normal neurologic examination except for cranial nerve abnormalities; normal neuroimaging (except for the accepted signs of raised intracranial pressure) including venography; a raised lumbar puncture opening pressure (≥25 cm CSF in adults and ≥28 cm CSF in children); and with normal CSF constituents. [22,23,24] This diagnosis would be coded by the administrative staff and transmitted to the hospital episode statistics.

To access information pertaining to all IIH admissions, validated International Classification of Diseases, Tenth Revision, Clinical Modification (ICD-10-CM) codes [26] and procedural classifications from the Office of Population, Censuses and Surveys Classification of Interventions and Procedures, 4th revision (OPCS-4) codes were used [27]. Whilst procedures and diagnoses are recorded within the HES database, medicines and lifestyle advice are not.

Exclusion criteria were applied to help refine the data and ensure against miscoding of secondary causes of raised intracranial pressure such as brain tumors, hydrocephalus and cerebral venous sinus thrombosis. Due to the very high number of admitted patients and comorbidities, we exclude those with a history of dialysis. These were likely to represent a secondary pseudotumor cerebri and the high admission rates could have potentially biased the results. Those who resided outside of England or whose residence was unknown were also excluded as accurate longitudinal patient tracking was not always possible (Figure 1).

### 2.2. Data Collection

Patient demographics on admission were recorded and included gender; ethnicity; geographical regional location, as classed by the Government Office Region (GOR); and social deprivation indices based on the English index of multiple deprivation (IMD) 2010. The IMD is the official measure of relative deprivation (for neighborhoods) in England and has been used frequently as a measure of relative deprivation to guide resource allocation and provision of services in the United Kingdom. Deprivation in this context refers to the relative disadvantage an individual experiences living in a certain area. The IMD is based on 38 routinely collected indicators, aggregated into seven weighted domains to represent different dimensions of deprivation, namely income, employment, health and disability, education and skills, barriers to housing and services, crime and environment. 

The IMD uses a small area-based model averaging 1500 people. Ranking the areas from 1 (most deprived area) to 32,844 (least deprived area), quintiles are calculated dividing the ranking into five equal groups [28].

Surgical treatments, including cerebrospinal fluid diversion procedures, optic nerve sheath decompression, cerebral venous sinus stenting, and bariatric surgery were recorded in this cohort. Numbers of admissions and revision rates of surgery were collected, along with 30 days readmission rates. Number of patients with visual impairment were recorded.

### 2.3. Statistical Analysis

The data were explored through descriptive analysis of variables.

### 2.4. Data Sharing Agreement

Proposals should be made to the corresponding author and reasonable requests will be granted access to the data. Requesters will be required to sign a data access agreement.

## 3. Results

### 3.1. Baseline Characteristics

HES identified 34,103 unique patients coded with IIH (ICD10=G932) between 1/1/2002 and 31/12/2019. To ensure that an unbiased representative population was analyzed, 5214 were excluded from the analysis (Figure 1).

The number of individual patients diagnosed with IIH was 28,794 during the study period. 4772 were male and 24,022 were female. The median age at diagnosis was 29 years (range: 22–40 years) with the peak age group at diagnosis was seen in females aged 25–29 years and in males under 13 years old (Figure 2A) (Table 1). In the general population, the incidence of IIH increased by 182% over the study period; in 2002 it was 1.84 per 100,000 rising to 5.18 per 100,000 in 2018/2019 (Figure 2B).

The patient’s socio-economic deprivation quintile (based on Index of Multiple Deprivation 2010) [28] was recorded, with the majority of cases (28%) residing in the most deprived areas (deprivation quintile 1; 8338 cases of IIH) and the least number of cases (12.5%) in areas of lower deprivation (deprivation quintile 5; 3709 cases of IIH) (Table 2).

### 3.2. Hospital Admissions

Between 2002 and 2019 there were a total of 104,315 hospital admissions for IIH, of which, for 55,525 (53%), IIH was the primary admission diagnosis (Figure 2C). Figure 2C shows a flattening of the curve of new diagnostic admissions around 2016 onwards. The majority of the cases had only one admission in the following year with an IIH diagnosis (62.7%); however, many of the cases had multiple admissions, and 1% of cases had in excess of 10 admissions (Table 3).

There was a reduction in repeat admissions, particular in the group that had over two admissions in the subsequent years following 2015/2016 (Table 4).

### 3.3. Management of IIH

The majority of cases (91.5%) were managed, as expected, without surgical procedure. As HES data codes diagnoses and procedures, no further analysis can be done, but it is presumed that these were medically managed with a combination of lifestyle advice regarding weight loss and medical therapy as recommended in national guidelines [22,23,24]. When surgery was performed the commonest procedure was a neurosurgical shunt (6.4%), followed by neovascular stent (0.99%), bariatric surgery (0.82%) and optic nerve sheath fenestration (0.53%) (Figure 2D). Over time the number of neurosurgical shunts has reduced (Figure 3A) and the percentage of the total IIH population requiring a neurosurgical shunt following a diagnosis of IIH has also decreased from 10.8% in 2002/2003 to 2.46% in 2018/2019 (Figure 3B). Records of neurovascular stenting for IIH above five cases per year were noted from 2008/2009 with 0.84% of cases treated with a stent, rising to a peak in 2013/2014 at 2.48% and falling to 0.61% 2018/2019 (Figure 3C).

### 3.4. Outcomes

The total number of neurosurgical revision surgery has fallen (Figure 4A). The percentage of revisions to primary shunt insertions per year has also reduced from 44.8% to 17.8% over the study period (Figure 4B). The mean 30 days emergency readmission rate for primary shunt over the total study period was 23.1%; however, when split down by year the trend rose to an all-time peak in 2008/2009 at 41% (Figure 4C). The mean 30 days emergency readmission rate was marginally higher for revision of shunt at 23.7%; however, a full breakdown was unable to be calculated due to masking of the data at source to ensure anonymization. The mean 30 days readmissions for bariatric surgery, neurovascular stent, and optic nerve sheath fenestration were 10.6%, 10.0% and 9.74% respectively.

Visual impairment decreased over the study period, with 3.26% of all cases in 2002/03, falling to 1.38% 2018/19, with a stepwise change seen in 2015/2016 (Figure 5).

## 4. Discussion

In this English population-based study of >104,000 hospitalizations for IIH, the incidence has continued to rise, with 5.18 per 100,000 having a new diagnosis in 2018/2019. This has predominately been considered to be due to the obesity epidemic [3,4,5]. Deprivation and social determinants are more commonly associated with IIH, which may contribute to the multiple hospital admissions recorded [29]. While the number of overall admissions (inpatient and day case) and emergency room attendances have risen, there has been a reduction in magnitude of multiple repeat admissions in the last four years of this observational study, with the key positive finding of a reduction in the rate of those coded with severe sight loss in IIH.

This data highlights that the number of primary surgical procedures are reducing in IIH. This could be due to an increased awareness of the condition and identification of more patients being correctly diagnosed that do not require surgical intervention. Another influence on this trend could have been the important recommendation by UK specific and European guidance publications that surgical procedures should be reserved for those with sight loss, and that headaches should be treated medically [23,24].

CSF shunts in IIH are known to be surgically more challenging compared to other shunts, with patient specific factors such as slit like ventricles and increased body habitus cited as contributing to this increased complexity [23]. Indeed, when compared to CSF shunts for all etiologies, shunts for IIH are associated with significantly higher revision rates [30,31]. Single-center studies report a wide range of revision rates in IIH shunts, ranging from 10% up to 60% [32,33,34,35]. In this study, the rising incidence of the disease and the better selection of patients for primary shunt have resulted in the reduction of patients, as a portion of the total number of IIH patients that had a shunt revision. Equally importantly, the proportion of shunt revisions to insertions has been declining, which is probably due to improving neurosurgical techniques such as image-guidance for ventricular cannulation, and developing shunt technology such as adjustable valves and anti-siphon devices. Setting correct indications for shunting and improvements in surgery have led to decreased neurosurgical morbidity.

The 30 days emergency readmission rate has been employed as a key performance indicator because readmission is costly and is a significant event for the individual. However, it has been challenged as a sole metric for gauging health care quality [36]. Factors which underlie the readmission rate are less well studied and there is a significant difference between the reported values across and within hospital specialties which appear to be largely driven by underlying pathology [36,37]. An example would be the wide range of readmission, between 12–21%, following acute ischemic stroke [38,39]. In neurosurgery complications following CSF, shunt surgeries are the commonest reason for early readmission, followed by surgical site infections and CSF leaks [40]. In a nationwide study in the USA, the 30-day readmission after ventricular shunting for all etiologies in adults was 16.5% [41]. In England over the study period, lumbo-peritoneal shunts were frequently used in some centers for IIH and are known to be associated with higher complication rates compared to ventricular shunts [42]. It was therefore not unexpected that the mean 30 days readmission rate was as high as 23% for the whole study period. What was unexpected was the rising 30-day readmission rate which was 41% in 2018/2019. What is not known is the combination of factors that contribute to readmission: A challenging surgery; adequate post-operative advice; clear pathway for post-admission advice; and overall education in a condition where patients have prior history of repeat admissions.

It is surprising that bariatric surgery was not performed more frequently in this population, given the association with weight gain and obesity driving the disease [10] and remission through weight loss [17]. Barriers in the UK to weight management surgery include the pathway to referrals from primary care, and limits on funding [10]. Bariatric surgery in established IIH has recently been reported as successful in terms of a significant sustained reduction of intracranial pressure and weight, as compared to Weight Watchers^TM^ in the setting of a randomized control trial [17]. Neurovascular stenting was not performed frequently in this population and this may be due to current uncontrolled clinical evidence and the lack of consensus on which patients benefit the most from this procedure [23,43]. Both methods of surgical intervention appear to have a more favorable 30 days emergency readmission and lower revision rate, as compared to shunts. However neurovascular stenting and other surgeries require further rigorous evaluation and comparison [43]. Optic nerve sheath fenestration is also uncommonly performed in England. This may be due to lack of surgical experience and local preferences. Retrospective chart reviews report favorable outcomes through the modern superior eyelid skin crease approach [44,45]. The English rate of fenestration appears to be a much lower than in the United States, where 1 in 10 surgically treated IIH cases had an optic nerve sheath fenestration, as compared to receiving a shunt [46].

The major strength of this study is the analysis of all IIH patients admitted along with patient care in English NHS hospitals over a 17 years period in a unique population-wide assessment. Using national data over this long time-period mitigates variation. Unlike that from insurance companies, this data is carefully curated and is unlikely to be biased due to variation in funding between hospitals, insurance coverage or the patients’ ability to pay for care. As in any databank, improvements in clinical record keeping and coding will improve the accuracy of the data.

Limitations of this databank study include uncertainty regarding the individual’s diagnosis and whether or not they fulfilled the recognized diagnostic criteria for IIH recommended by national and international guidance [22,23,24]. It is well known that there is overdiagnosis of IIH [47], therefore there is the risk that the numbers could be over reported. When compared to the primary care data in the UK, which have previously been published, the incidence over corresponding years is similar. [4] Likewise, within this cohort there are patients over the age of 65 years (Figure 2A), which is not typical of the known phenotype of IIH, described by the IIH literature as a disease that occurs with the major risk factor being weight gain in women of working age [1,2,7]. A recent retrospective case series analysis of IIH above the age of 65 years reported that the diagnosis was more commonly seen in males, with incidentally found papilloedema; they had fewer headaches, and on investigation had lower lumbar puncture opening pressure, as compared to IIH controls below the age of 50 years [48]. These older age group cases may represent truly “idiopathic” cases with raised intracranial pressure of unknown etiology, and further research should be done to define this disease sub-type in this age group. 

Similarly, within this analysis there is a high portion of males coded with IIH (Figure 2A). Prior to modern imaging studies the portion of males was reported as high as 50% of case series [49]. Following MRI introduction this was refined to between 8–19% [50,51,52]. In the recent literature, despite there being a wide range in the proportions of males diagnosed with IIH of between 9% to 27%, [53,54] it should be borne in mind that, within two specialist IIH clinics who compared their clinical data in the USA and UK, the proportion of males diagnosed with the condition was much lower at 6% and 4.1%, respectively [55]. This may reflect overdiagnosis of the condition in males. Whilst overall in this analysis the portion of males was 17%, it should be recognized that this took in all age groups and the trends in childhood do reflect the literature regarding pediatric IIH where pre-puberty age groups have similar portions of male to female children [56].

Further limitations of this data are that HES does not include medical treatments and medications; measurements of body mass index; or smoking status, although it does include co-morbid illness coding. This not a requirement for reporting in each case, therefore it is likely that comorbid diseases are underreported, hence why no further analysis of the HES data was undertaken because of the risk of inaccurate reporting bias. 

Changing awareness of the condition due to publication of the IIH guidelines [23,24] and emerging evidence [1,2] could have influenced the trends, in particular, the recommendation that CSF shunting should only be performed for sight threatening disease and not headache symptoms. Similarly, recommendations for medical headache management could have possibly reduced the number of those who had previously been admitted for repeat lumbar punctures or sought surgery for management. Better patient selection and improvements in shunt surgery have likely led to a reduction in shunt revisions. 

A key area for future exploration is the 30-day emergency readmission rate for surgical procedures, particularly shunting. This is a key performance indicator for the NHS, and as such is a robustly measured metric. With further granular data, clinicians and policy makers would be able to develop strategies to address the rising trend of readmissions, and how shunt technology may be able to address surgical challenges in IIH. Further observation of these trends is important, to monitor the likely changes such observed following national lockdown and the Covid-19 pandemic era, when it appears neurosurgical shunts may have had increased utilization of services [9].

## 5. Conclusions

This increasing incidence and number of hospital admissions needs to be addressed and, although the wider social demographic is likely to be more complex, change, communication and effective treatments may be more achievable in the short to medium term. This condition requires multidisciplinary working to improve patient morbidity from repeat admissions, emergency readmission following surgery and sight loss.

## Figures and Tables

**Figure 1 life-11-00417-f001:**
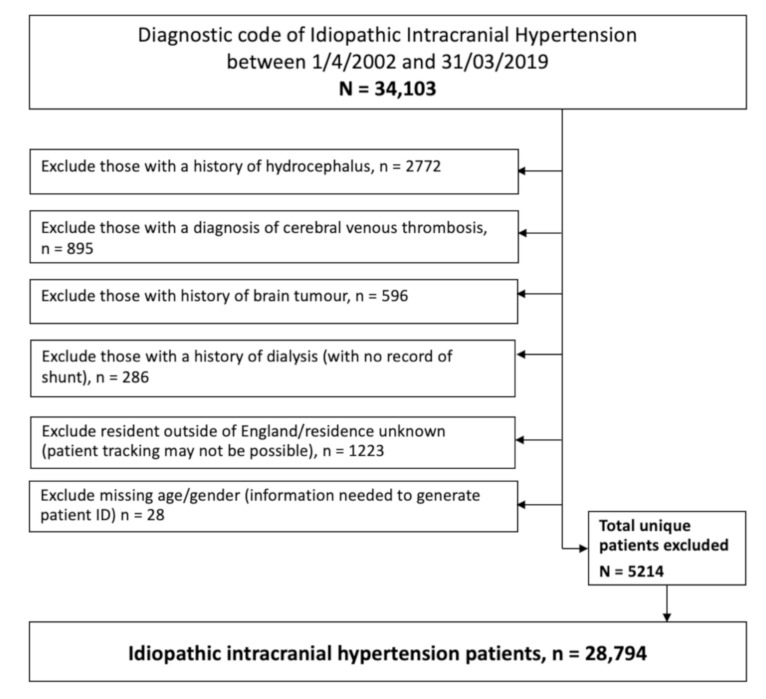
Consort diagram detailing exclusion criteria.

**Figure 2 life-11-00417-f002:**
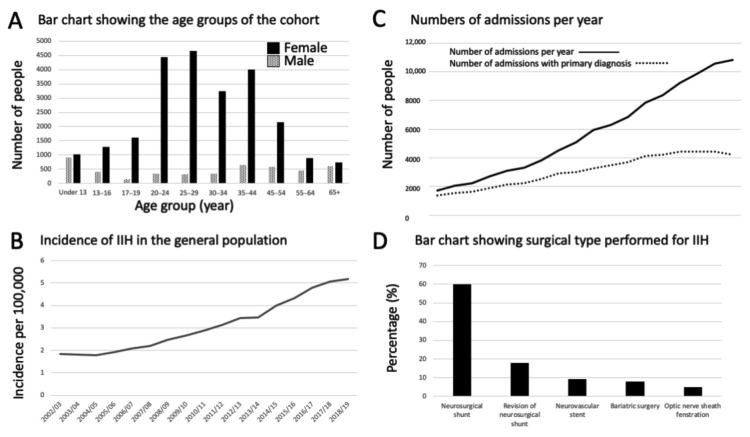
(**A**) Number of IIH cases as determined by age group and sex for the total study period; (**B**) Incidence of new diagnosis of IIH per 100,000 over the study period; (**C**) Number of all admissions per year, with the number of incident IIH admissions; and (**D**) the percentage of surgical procedure type performed in IIH.

**Figure 3 life-11-00417-f003:**
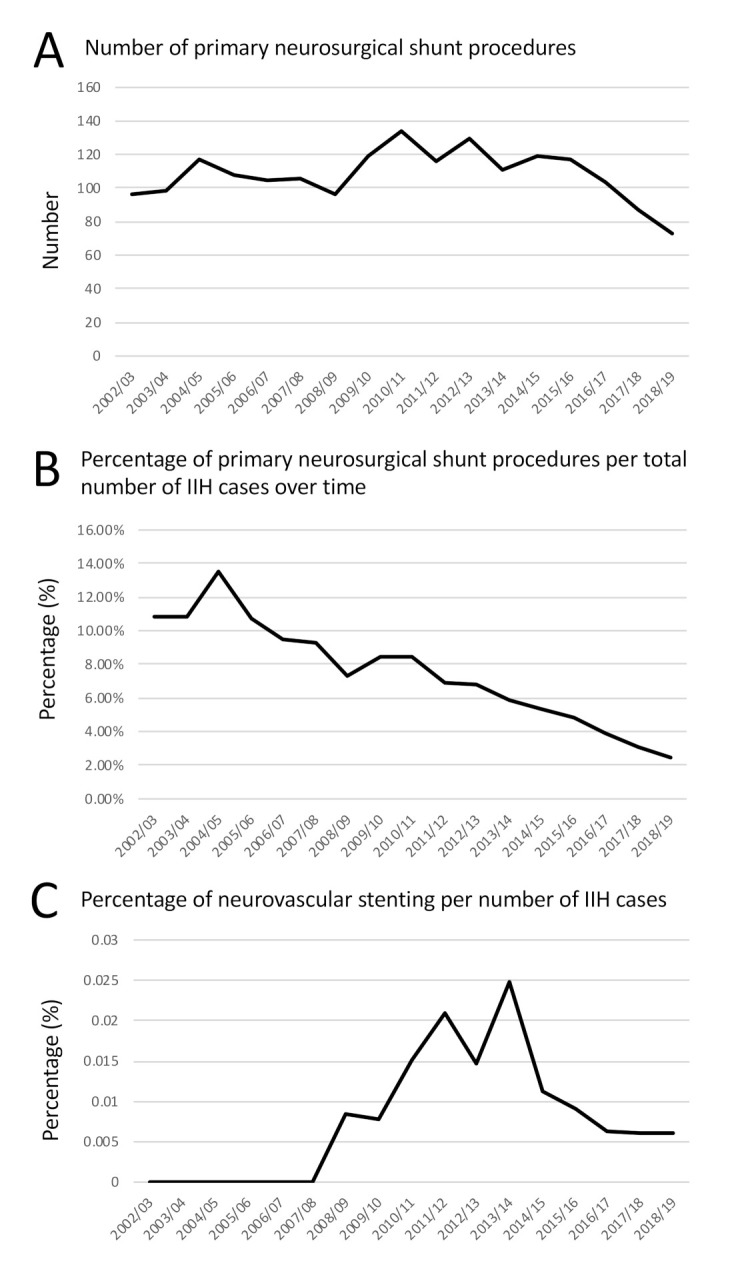
Primary shunts and stents in IIH. (**A**) Number of primary neurosurgical shunts performed over time (**B**) Line graph shows the percentage of primary neurosurgical shunt procedures per total number of IIH cases diagnosed in the same year over time; (**C**) Line graph shows the percentage of neurovascular stenting procedures per total number of IIH cases diagnosed in the same year over time.

**Figure 4 life-11-00417-f004:**
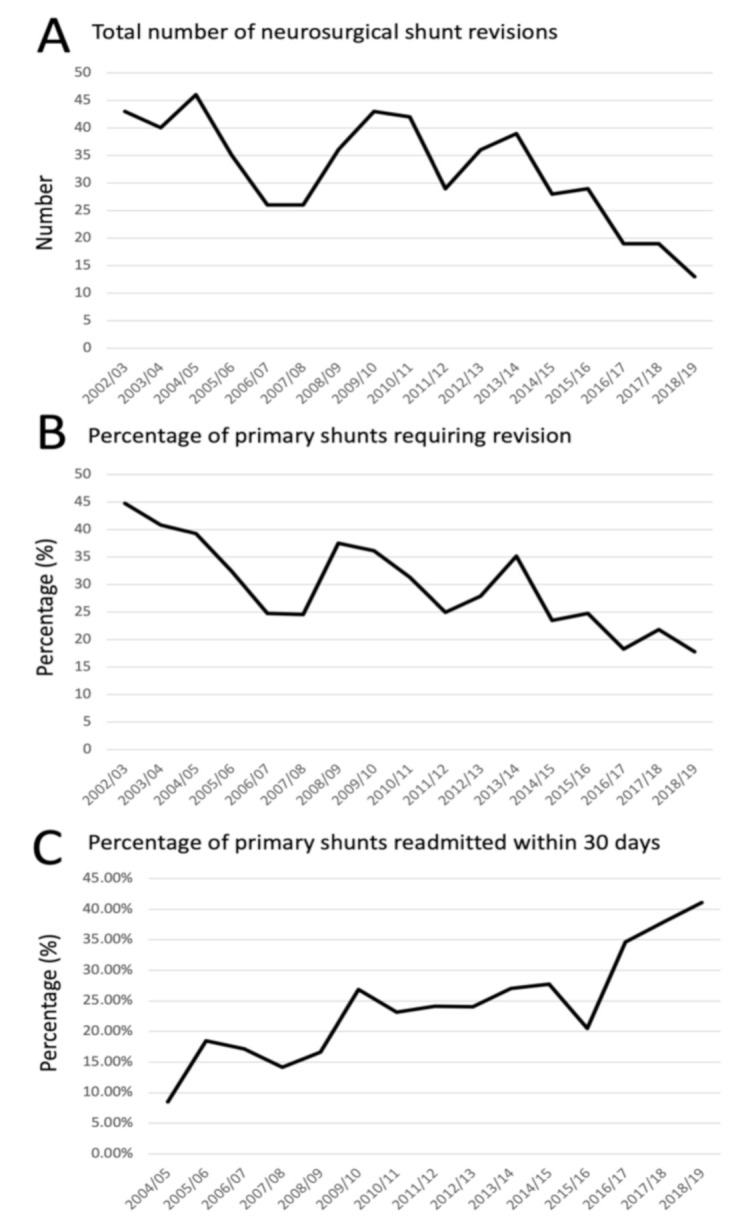
Morbidity as evidenced by revisions and 30 days readmission rates for shunts. (**A**) The total number of neurosurgical shunt revisions performed over time; (**B**) the percentage of primary shunt requiring revision per year; (**C**) the percentage of shunts readmitted within 30 days.

**Figure 5 life-11-00417-f005:**
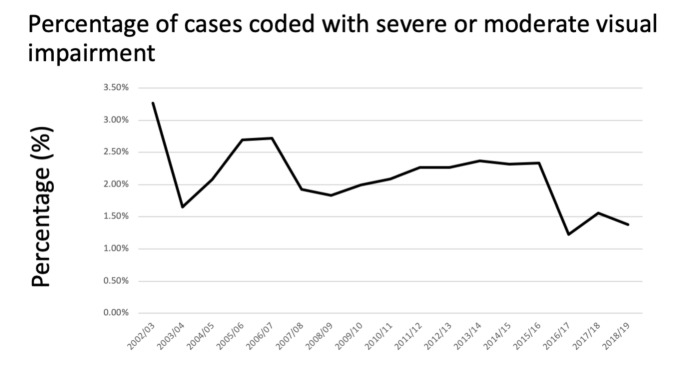
Reduction in number of cases coded for visual impairment. (These codes included binocular or monocular blindness; severe binocular or monocular visual impairment and moderate binocular or monocular visual impairment.).

**Table 1 life-11-00417-t001:** The age group of the IIH cohort (2002–2019).

Age (Years)	Males (Number (%))	Females (Number (%))	All People (Number (%))
Under 13	907 (19.01%)	1023 (4.26%)	1930 (6.68%)
13–16	403 (8.45%)	1278 (5.32%)	1681 (5.82%)
17–19	154 (3.23%)	1604 (6.68%)	1758 (6.09%)
20–24	349 (7.31%)	4444 (18.50%)	4793 (16.59%)
25–29	320 (6.71%)	4649 (19.35%)	4969 (17.20%)
30–34	339 (7.10%)	3238 (13.48%)	3577 (12.38%)
35–44	642 (13.45%)	4010 (16.69%)	4652 (16.10%)
45–54	588 (12.32%)	2146 (8.93%)	2734 (9.46%)
55–64	461 (9.66%)	899 (3.74%)	1360 (4.71%)
65+	609 (12.76%)	731 (3.04%)	1340 (4.64%)
Total Number	4772	24,022	28,794

**Table 2 life-11-00417-t002:** Deprivation based on the Index of Multiple Deprivation 2010 [28].

Deprivation Quintile	Males	Females	Total
1—Most Deprived	1304 (27.33%)	7034 (29.28%)	8338 (27.99%)
2	1003 (21.02%)	5698 (23.72%)	6701 (22.49%)
3	928 (19.45%)	4376 (18.22%)	5304 (17.80%)
4	773 (16.20%)	3669 (15.27%)	4442 (14.91%)
5—Least Deprived	708 (14.84%)	3001 (12.49%)	3709 (12.45%)
Unknown	56 (1.17%)	244 (1.02%)	300 (1.01%)

**Table 3 life-11-00417-t003:** Number of additional admissions following the diagnostic episode.

Number of Additional Admissions in the First Year of Diagnosis	Number of Patients (% of Total Number of Patients)
0	18,046 (62.67%)
1	5221 (18.13%)
2	2352 (8.17%)
3	1143 (3.97%)
4	665 (2.31%)
5	420 (1.46%)
6	309 (1.07%)
7	199 (0.69%)
8	116 (0.40%)
9	86 (0.30%)
10+	237 (0.82%)

**Table 4 life-11-00417-t004:** Number of subsequent admissions following the diagnostic episode.

Diagnosis Year	Number of Subsequent Admissions Following Diagnosis					
	0		1 to 2		More than 2	
2002/03	554	62.32%	220	24.75%	115	12.94%
2003/04	581	64.06%	222	24.48%	104	11.47%
2004/05	527	60.92%	228	26.36%	110	12.72%
2005/06	620	61.75%	261	26.00%	123	12.25%
2006/07	680	61.71%	295	26.77%	127	11.52%
2007/08	703	61.67%	301	26.40%	136	11.93%
2008/09	802	61.31%	351	26.83%	155	11.85%
2009/10	837	59.57%	385	27.40%	183	13.02%
2010/11	994	62.83%	399	25.22%	189	11.95%
2011/12	1031	61.48%	438	26.12%	208	12.40%
2012/13	1171	61.73%	507	26.73%	219	11.54%
2013/14	1184	62.45%	498	26.27%	214	11.29%
2014/15	1395	62.28%	585	26.12%	260	11.61%
2015/16	1468	61.27%	673	28.09%	255	10.64%
2016/17	1727	64.13%	697	25.88%	269	9.99%
2017/18	1823	64.55%	742	26.27%	259	9.17%
2018/19	1949	65.64%	771	25.97%	249	8.39%

## Data Availability

Proposals should be made to the corresponding author and reasonable requests will be granted access to the data. Requesters will be required to sign a data access agreement.

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
