# Peer review of "Idiopathic Intracranial Hypertension: Evaluation of Admissions and Emergency Readmissions through the Hospital Episode Statistic Dataset between 2002–2020"

_life, 2021, doi:10.3390/life11050417_

Round 1

Reviewer 1 Report

The authors presented an original study focused on admissions, treatment options and emergency readmissions associated with cases of Idiopathic intracranial hypertension (IIH), based on nationwide Hospital Episode Statistics from England. Despite some strength of the manuscript, specifically analysis covering a vast number of cases, it has some issues which need to be addressed:

- Please, explain in more details the process of IIH clinical diagnosis. Is it only based on patient complaints and some basic physical examinations, or is it related to e.g. neuroimaging or other highly specialized diagnostic techniques?

- It is very unclear what were the main therapeutic strategies applied to the majority of patients. If more that 91% of IIH cases were managed without neurosurgical procedures, then what kind of interventions were proceeded? There is no information in this area.

- It is absolutely necessary to add more typically clinical characteristics of the collected sample. Authors presented only age and sex, what says nothing about the specificity of the analyzed disease / medical problem. Please, add data such as weight, BMI, co-morbidities, psychoactive substances misuse, vascular health indicators, percentage of patients after any form of traumatic brain injury, number of hospitalizations due to other reasons than IIH, percentage of patients smoking, with diabetes, dementia.

- Why authors decide to use only the most basic descriptive statistics having such interesting and valuable data? Maybe it would be worth analyzing at least relationships between clinical characteristics of IIV and obesity?

- There is almost no discussion regarding biomedical mechanisms linking obesity with IIH incidents, although such links are suggested several times throughout the manuscript.

- There are a lot of flaws regarding formal aspects of the manuscript: lack of captions of Figures 1 and 2, bad formation of Table 1, the results depicted in the Figures are insufficiently discussed and interpreted in the text of the manuscript, there should be numbered subsections in the 3. Results section, references in the text should be adjusted to editorial requirements.

Author Response

We would like to thank the reviewer for their time spent on our manuscript and hope they agree with the additions and corrections following their recommendations, alongside attending to the other two reviewer comments.

The authors presented an original study focused on admissions, treatment options and emergency readmissions associated with cases of Idiopathic intracranial hypertension (IIH), based on nationwide Hospital Episode Statistics from England. Despite some strength of the manuscript, specifically analysis covering a vast number of cases, it has some issues which need to be addressed:

- Please, explain in more details the process of IIH clinical diagnosis. Is it only based on patient complaints and some basic physical examinations, or is it related to e.g. neuroimaging or other highly specialized diagnostic techniques?

Added to the methods:

A diagnosis of IIH will be made by the hospital specialist.  Typically in the UK hospital specialists should follow the guidelines for this diagnosis which include the presence of papilloedema; normal neurologic examination except for cranial nerve abnormalities; normal neuroimaging (except for the accepted signs of raised intracranial pressure)including venography; a raised lumbar puncture opening pressure (≥25cm CSF in adults and ≥28cm CSF in children; and with normal CSF constituents. [22][23][24] This diagnosis would be coded by the administrative staff and transmitted to the hospital episode statistic.

- It is very unclear what were the main therapeutic strategies applied to the majority of patients. If more that 91% of IIH cases were managed without neurosurgical procedures, then what kind of interventions were proceeded? There is no information in this area.

They are usually advised to lose weight (as this is the mainstay of treatment and is discussed in the introduction) and for some are given a carbonic anhydrase inhibitor.  Advising weight loss and drugs are not recorded in this health care database  To make this clearer we have added the following:

To access information pertaining to all IIH admissions, validated International Classification of Diseases, Tenth Revision, Clinical Modification (ICD-10-CM) codes[26] and procedural classifications from the Office of Population, Censuses and Surveys Classification of Interventions and Procedures, 4th revision (OPCS-4) codes were used [27]. Whilst procedures and diagnoses are recorded within the HES database, medicines and lifestyle advice are not.

Within the results we have added:

3.3 Management of IIH

The majority of cases (91.5%) were managed, as expected, without surgical procedure. As HES data codes diagnoses and procedures no further analysis can be done but it is presumed these were medical managed with a combination of lifestyle advice regarding weight loss and medical therapy as recommended in national guidelines.[22][23][24]

- It is absolutely necessary to add more typically clinical characteristics of the collected sample. Authors presented only age and sex, what says nothing about the specificity of the analyzed disease / medical problem. Please, add data such as weight, BMI, co-morbidities, psychoactive substances misuse, vascular health indicators, percentage of patients after any form of traumatic brain injury, number of hospitalizations due to other reasons than IIH, percentage of patients smoking, with diabetes, dementia.

The hospital episode statistic data does not record body mass index, or smoking status.  It does record co-morbidities, but they are not deemed reliable enough to be used for analysis.  We have added to the limitation section of the discussion the following:

No further analysis of the HES data was not undertaken because of the risk of inaccurate reporting bias. A limitation of this data is that HES does not include medical treatments and medications; measurements of body mass index; or smoking status. Although it does include co-morbid illness coding this is not a requirement for reporting in each case, therefore it is likely that comorbid diseases are underreported and hence why they have not been analyzed here.

- Why authors decide to use only the most basic descriptive statistics having such interesting and valuable data? Maybe it would be worth analyzing at least relationships between clinical characteristics of IIV and obesity?

As discussed in the paragraph above, we do not have any further clinical characteristics and BMI data. We are in fact working on primary care data which contains BMI and medicines/drugs but this database is not linked (yet) in the UK. 

- There is almost no discussion regarding biomedical mechanisms linking obesity with IIH incidents, although such links are suggested several times throughout the manuscript.

As discussed in the paragraph above, we do not have the BMI data. This is similar to other national coding data (for example the Swedish registry).  As noted above we are working on a further case controlled nested on primary care data which contains BMI and medicines/drugs but this database is not linked (yet) in the UK.  Our previous work highlights this:

  1. Adderley NJ, et al. Association between idiopathic intracranial hypertension and risk of cardiovascular diseases in women in the United Kingdom. JAMA Neurol. 2019;76:1088–1098

- There are a lot of flaws regarding formal aspects of the manuscript:

lack of captions of Figures 1 and 2, corrected/added.

bad formation of Table 1, I did ask the Life team to help with this table formatting, as the template keeps reformatting it incorrectly.

the results depicted in the Figures are insufficiently discussed and interpreted in the text of the manuscript, We have highlighted the figures in the results and discussed them in the discussion.

Limitations of this databank study include uncertainty regarding the individual’s diagnosis and that they fulfilled the recognized diagnostic criteria for IIH that have been recommended by national and international guidance.[22][23][24] It is well known that there is overdiagnosis of IIH [47], therefore there is the risk that the numbers could be over reported.  When compared to the primary care data in the UK, which have previously been published the incidence over corresponding years is similar. [4] Likewise within this cohort there are patients over the age of 65 years [Figure 2A], this is not typical of the known phenotype of IIH as described by the IIH literature as a disease that occurs with the major risk factor being weight gain in women of working age.[1][2][7] A recent retrospective case series analysis of diagnosed with IIH above the age of 65 years reported that the diagnosis was more commonly seen in males, with incidentally found papilloedema; they had fewer headaches, and on investigation had lower lumbar puncture opening pressure, as compared to IIH controls below the age of 50 years.[48] These older age group cases may represent truly “idiopathic” cases with raised intracranial pressure of unknown etiology, and further research should be done to define this disease sub-type in this age group.

Similarly, within this analysis there is a high portion of males coded with IIH [Figure 2A]. Prior to modern imaging studies the portion of males was reported as high as 50% of case series.[49] Following MRI introduction this was refined to between 8-19%.[50][51][52][53] In the recent literature despite there being a wide range of portions of males diagnosed with IIH between 9% to 27%, [54][55] it should be borne in mind that within two specialist IIH clinics who compared their clinical data in the USA and UK, the portion of males diagnosed with the condition was much lower at 6% and 4.1%, respectively.[56] This may reflect overdiagnosis of the condition in males. Whilst overall in this analysis the portion of males was 17%, it should be recognized this analysis took in all age groups and the trends in childhood do reflect the literature regarding pediatric IIH where the pre-puberty age groups have similar portions of male to female children.[57]

there should be numbered subsections in the 3. Added

Results section, references in the text should be adjusted to editorial requirements. We will work with the Life team to address the journal requirements.

Reviewer 2 Report

The authors describe the characteristics and re-admission rates of IIH patients in England. They used the Hospital Episode Statistics database. They provide information on the type of surgical interventions used, readmission rates.    Some comments:   Methods Can the authors please clarify how confident they were that clinical codes actually retrieved patients with IIH. Is there any reference or support that can be cited that previously looked at this? Has this been previously validated? 
There is a big problem with the overdiagnosis of IIH (See Fisayo et al. Neurology). Can the authors also discuss how many neuro-ophtahlmologists are in the UK and what specialties made this diagnosis in these cases. Did all patients have an eye exam?   Do all patients with IIH typically have a hospital admission in the UK? From what I understand, most IIH patients are not admitted to hospital for workup and this is performed as an outpatient. I assume these patients were not captured by the data set?
Can the authors comment on why roughly 17% of IIH patients were men? This seems to be larger that other studies
Can the authors please also discuss why there is such a large number of patients over 65 being diagnosed with IIH. How does this compare to the literature?   Is there any data on how many patients were prescribed medications for treatment of IIH?

Author Response

We would like to thank the reviewer for their time spent on our manuscript and hope they agree with the additions and corrections following their recommendations, alongside attending to the other two reviewer comments.

The authors describe the characteristics and re-admission rates of IIH patients in England. They used the Hospital Episode Statistics database. They provide information on the type of surgical interventions used, readmission rates.    Some comments:   Methods Can the authors please clarify how confident they were that clinical codes actually retrieved patients with IIH.

  • There have been many publications using HES data, and whilst we have compared the high level numbers with our prior publication and they appear to correlate well. The Adderley et al is a primary care database in UK, and HES secondary care : Adderley NJ, et al. Association between idiopathic intracranial hypertension and risk of cardiovascular diseases in women in the United Kingdom. JAMA Neurol. 2019;76:1088–1098

Is there any reference or support that can be cited that previously looked at this?

Has this been previously validated?  

  • HES data has been validated by the primary care CPRD data in a number of publications such as Burkard T, Rauch M, Jick SS, Meier CR. Validity of bariatric surgery codes in the UK Clinical Practice Research Datalink (CPRD) GOLD compared with Hospital Episodes Statistics. Pharmacoepidemiol Drug Saf. 2021 Mar 6. doi: 10.1002/pds.5221. What we cannot attest to validity in IIH.

There is a big problem with the overdiagnosis of IIH (See Fisayo et al. Neurology).

  • We agree and whilst this is recognised, essential all these patients were coded, and likely treated as IIH. We have added to the limitations:

Also, it would be uncertain if each of the individuals fulfilled the recognized diagnostic criteria for IIH. It is well known that there is overdiagnosis of IIH.[47]

Can the authors also discuss how many neuro-ophtahlmologists are in the UK and what specialties made this diagnosis in these cases.

  • The HES does not allow us to drill down to this type of data. In the UK we have circa 75 people with a declared interest in neuro-ophthalmology.  The diagnosis will be made by neurology, neuro-op and neuro-surgery.

Did all patients have an eye exam?  

  • HES does not record this. However the Association of British Neurologist guidelines have been popularised both within ophthalmology and neurosurgery, so it would be classed as a clinical negligence case if they had not had papilloedema confirmed.

Do all patients with IIH typically have a hospital admission in the UK? From what I understand, most IIH patients are not admitted to hospital for workup and this is performed as an outpatient. I assume these patients were not captured by the data set? 

  • HES codes admissions, as accident and emergency attendances, ambulatory care (where LPs are usually performed from outpatients) and admissions to the ward. Few cases would be missed as most patient identified with papilloedema go to either eye casualty or main accident and emergency; if they went directly to neurology, ophthalmology or neurosurgery the majority would have a Lumbar puncture done and coded as a procedure on ambulatory care or as a day case or within inpatient wards. We have added to the methods:

This includes accident and emergency attendances, admitted hospital care (inpatient activity) and ambulatory care admissions (day case admissions).

Can the authors comment on why roughly 17% of IIH patients were men? This seems to be larger that other studies 

  • We have added the following

Similarly, within this analysis there is a high portion of males coded with IIH [Figure 2A]. Prior to modern imaging studies the portion of males was reported as high as 50% of case series.[49] Following MRI introduction this was refined to between 8-19%.[50][51][52][53] In the recent literature despite there being a wide range of portions of males diagnosed with IIH between 9% to 27%, [54][55] it should be borne in mind that within two specialist IIH clinics who compared their clinical data in the USA and UK, the portion of males diagnosed with the condition was much lower at 6% and 4.1%, respectively.[56] This may reflect overdiagnosis of the condition in males. Whilst overall in this analysis the portion of males was 17%, it should be recognized this analysis took in all age groups and the trends in childhood do reflect the literature regarding pediatric IIH where the pre-puberty age groups have similar portions of male to female children.[57]

Can the authors please also discuss why there is such a large number of patients over 65 being diagnosed with IIH. How does this compare to the literature?  

  • We have added the following

Likewise within this cohort there are patients over the age of 65 years [Figure 2A], this is not typical of the known phenotype of IIH as described by the IIH literature as a disease that occurs with the major risk factor being weight gain in women of working age.[1][2][7] A recent retrospective case series analysis of diagnosed with IIH above the age of 65 years reported that the diagnosis was more commonly seen in males, with incidentally found papilloedema; they had fewer headaches, and on investigation had lower lumbar puncture opening pressure, as compared to IIH controls below the age of 50 years.[48] These older age group cases may represent truly “idiopathic” cases with raised intracranial pressure of unknown etiology, and further research should be done to define this disease sub-type in this age group.

Is there any data on how many patients were prescribed medications for treatment of IIH?

  • Sadly the HES does not code medical treatment, we have added:

To access information pertaining to all IIH admissions, validated International Classification of Diseases, Tenth Revision, Clinical Modification (ICD-10-CM) codes[26] and procedural classifications from the Office of Population, Censuses and Surveys Classification of Interventions and Procedures, 4th revision (OPCS-4) codes were used [27]. Whilst procedures and diagnoses are recorded within the HES database, medicines and lifestyle advice are not.

  • Within the results we have added:

3.3 Management of IIH

The majority of cases (91.5%) were managed, as expected, without surgical procedure. As HES data codes diagnoses and procedures no further analysis can be done but it is presumed these were medical managed with a combination of lifestyle advice regarding weight loss and medical therapy as recommended in national guidelines.[22][23][24]

Reviewer 3 Report

This is an interesting article evaluating changes in hospital admissions and surgeries for Idiopathic intracranial hypertension, and efforts on such an important topic should be lauded; however, very important aspects in this manuscript need to be clarified and a major revision is needed before possible evaluation for publication.

As the Authors themselves have recently written papers on updates on diagnosis and management of idiopathic intracranial hypertension, and criteria for the diagnosis of the various causes of intracranial hypertension have been amended to more clearly separate IIH from secondary causes, the choice of including ICD-10-CM IIH diagnosis since 2002 (almost 20 years ago) may lead to a bias in the some of the conclusions of the present manuscript.

In fact, it might be hard to support this paper’s results starting from definitions which have been modified during the years.

I would suggest to take into consideration a shorter (and more recent) period of observation, or separating the results in two different time periods: before and after proposed updated diagnostic criteria to incorporate advances and insights into the disorder realized over the past 10 years.

The study limitations are not clearly stated; it is said that “it would be uncertain if each of the individuals fulfilled the recognised diagnostic criteria for IIH”, and this represents a major bias of this study, therefore potentially leading to wrong assumptions.

The criteria used for clinical definition of “severe visual impairment” must be clearly stated.

In the Results section it is said that “the least number of cases (12.5%) (resides) in areas of lower deprivation (deprivation quintile 5; 3001 cases of IIH)”, I do believe that the last number should be 3709, that is the total number of cases, and not 3001 (just the females).

In the first lines of the Discussion section it is said that IIH incidence is “5.8 per 100,000”; I do believe there is a mistake, as previously in the Results section it is said “rising to 5.18 per 100,000 in 2018/2019”.

Please rephrase: “over 2 admissions FOLLOWING diagnosis FOLLOWING 2015/2016”.

Please remove, in the Author Contributions section: “For research articles with several authors, a short paragraph specifying their individual contributions must be provided. The following statements should be used”

Author Response

Reviewer 3

We would like to thank the reviewer for their time spent on our manuscript and hope they agree with the additions and corrections following their recommendations, alongside attending to the other two reviewer comments.

This is an interesting article evaluating changes in hospital admissions and surgeries for Idiopathic intracranial hypertension, and efforts on such an important topic should be lauded; however, very important aspects in this manuscript need to be clarified and a major revision is needed before possible evaluation for publication.

As the Authors themselves have recently written papers on updates on diagnosis and management of idiopathic intracranial hypertension, and criteria for the diagnosis of the various causes of intracranial hypertension have been amended to more clearly separate IIH from secondary causes, the choice of including ICD-10-CM IIH diagnosis since 2002 (almost 20 years ago) may lead to a bias in the some of the conclusions of the present manuscript.

In fact, it might be hard to support this paper’s results starting from definitions which have been modified during the years.

I would suggest to take into consideration a shorter (and more recent) period of observation, or separating the results in two different time periods: before and after proposed updated diagnostic criteria to incorporate advances and insights into the disorder realized over the past 10 years.

We thank you for your suggestion, the other two reviewers have not voiced this concern.  We have discussed this suggestion as a team, and wish to keep the results as they represent ‘real life’ coding with the national health service as we have moved over the 17 year period. 

In the UK we have had a stepwise change in the publicity of IIH particularly during the periods of surveys with the Association of British Neurologists for our IIH recommendations; the James Lind Alliance Priority Setting Partnership with all disciplines involved with patients and carers.  Our group’s feeling is that whilst the diagnostic criteria publication may have influenced specialists, the wider publications we have worked and presented at annual congresses (RCOphth, ABN and SBNS) have influenced the general neurology doctors and neurosurgeons, who would be diagnosing this condition.

The study limitations are not clearly stated; it is said that “it would be uncertain if each of the individuals fulfilled the recognised diagnostic criteria for IIH”, and this represents a major bias of this study, therefore potentially leading to wrong assumptions.

We have enhanced the methods section to highlight the challenges of nationally held databases. 

We have changed the limitation section to include the following comparisons to the literature:

Limitations of this databank study include uncertainty regarding the individual’s diagnosis and that they fulfilled the recognized diagnostic criteria for IIH that have been recommended by national and international guidance.[22][23][24] It is well known that there is overdiagnosis of IIH [47], therefore there is the risk that the numbers could be over reported.  When compared to the primary care data in the UK, which have previously been published the incidence over corresponding years is similar. [4] Likewise within this cohort there are patients over the age of 65 years [Figure 2A], this is not typical of the known phenotype of IIH as described by the IIH literature as a disease that occurs with the major risk factor being weight gain in women of working age.[1][2][7] A recent retrospective case series analysis of diagnosed with IIH above the age of 65 years reported that the diagnosis was more commonly seen in males, with incidentally found papilloedema; they had fewer headaches, and on investigation had lower lumbar puncture opening pressure, as compared to IIH controls below the age of 50 years.[48] These older age group cases may represent truly “idiopathic” cases with raised intracranial pressure of unknown etiology, and further research should be done to define this disease sub-type in this age group.

Similarly, within this analysis there is a high portion of males coded with IIH [Figure 2A]. Prior to modern imaging studies the portion of males was reported as high as 50% of case series.[49] Following MRI introduction this was refined to between 8-19%.[50][51][52][53] In the recent literature despite there being a wide range of portions of males diagnosed with IIH between 9% to 27%, [54][55] it should be borne in mind that within two specialist IIH clinics who compared their clinical data in the USA and UK, the portion of males diagnosed with the condition was much lower at 6% and 4.1%, respectively.[56] This may reflect overdiagnosis of the condition in males. Whilst overall in this analysis the portion of males was 17%, it should be recognized this analysis took in all age groups and the trends in childhood do reflect the literature regarding pediatric IIH where the pre-puberty age groups have similar portions of male to female children.[57]

The criteria used for clinical definition of “severe visual impairment” must be clearly stated.

HES codes we used for inclusion were:

H540 - Blindness, binocular

H541 - Severe visual impairment, binocular

H542 - Moderate visual impairment, binocular

H544 - Blindness, monocular

H545 - Severe visual impairment, monocular

H546 - Moderate visual impairment, monocular

And for exclusion were

H53.9 visual disturbance, unspecified

H470 - Disorders of optic nerve, not elsewhere classified

H543 - Mild or no visual impairment, binocular

Whilst we cannot denote the exact amount of visual impairment clinically, we used these codes to allow us to look at trends.  We have annotated at the figure legend to make this transparent.

Figure 5: Reduction in number of cases coded for visual impairment (These codes included binocular or monocular blindness; severe binocular or monocular visual impairment and moderate binocular or monocular visual impairment.)

In the Results section it is said that “the least number of cases (12.5%) (resides) in areas of lower deprivation (deprivation quintile 5; 3001 cases of IIH)”, I do believe that the last number should be 3709, that is the total number of cases, and not 3001 (just the females).

Thank you for picking this typo up. We do appreciate this accuracy.

In the first lines of the Discussion section it is said that IIH incidence is “5.8 per 100,000”; I do believe there is a mistake, as previously in the Results section it is said “rising to 5.18 per 100,000 in 2018/2019”.

Thank you for picking this typo up. We do appreciate this accuracy.

Please rephrase: “over 2 admissions FOLLOWING diagnosis FOLLOWING 2015/2016”.

There was a reduction in repeat admissions, particular in the group that had over 2 admissions in the subsequent years following 2015/2016 [Table 4].

Please remove, in the Author Contributions section: “For research articles with several authors, a short paragraph specifying their individual contributions must be provided. The following statements should be used”

Removed.

Round 2

Reviewer 1 Report

The authors addressed all suggestions, although to some extent they did not make significant changes due to the lack of access to specific data and registers.

Reviewer 3 Report

The Authors have addressed all my concerns, no further changes are suggested.